# Detecting Land Surface Temperature Variations Using Earth Observation at the Holy Sites in Makkah, Saudi Arabia

Ahmad Fallatah [1,2,*] and Ayman Imam [3]

1 Department of Geomatics, Faculty of Architecture and Planning, King Abdulaziz University, Jeddah 21589, Saudi Arabia

2 The Center of Excellence in Smart Environment Research, King Abdulaziz University, Jeddah 21589, Saudi Arabia

3 Department of Urban and Regional Planning, Faculty of Architecture and Planning, King Abdulaziz University, Jeddah 80200, Saudi Arabia; aeimam@kau.edu.sa

* Correspondence: amfallatah@kau.edu.sa; Tel.: +966-506622399

**Abstract:** During Hajj, Muslims throughout the globe assemble at the holy sites in Makkah, Saudi Arabia. The Saudi government aims to increase the number of pilgrims. To ensure the pilgrims' safety from the impact of surface urban heat island (SUHI), a scientific approach using artificial intelligence and Earth observation (EO) is recommended for crowd management and human health. SUHI is usually measured using satellite LST data. UHIs impact the walkability of cities in hot climates. The development of digital technologies has provided researchers with a better understanding of crowd management modeling to control such a mass gathering, especially within the summer season and in drought regions. In this study, an approach was used to detect the UHI in holy sites and understand the factors causing them. To achieve this goal, EO data were used to calculate the LST using the Landsat 8 thermal band (TIRS) and calculating the surface emissivity and Normalized Difference Vegetation Index (NDVI). Then, UHIs were identified by adding the mean of the LST to half of its standard deviation. Based on the results of this study, LST had a strong correlation with NDVI (negative) in Arafah. In addition, the strength of the correlation became much weaker within Mina and Muzdalifah. As for the correlation of LST and elevation, the strength appeared to be weak but negative in Arafah, but stronger in Muzdalifah and Mina. The results show that there is a certain correlation between LST, NDVI, and NDBI and elevation in the study area. Using Earth observation technologies can help in studying climate change.

**Keywords:** urban heat island (UHI); land surface temperature (LST); Hajj; holy sites; crowd management; pedestrian network; Earth observation

## 1. Introduction

The urban heat island (UHI) is a phenomenon that is commonly witnessed across the world [1,2]. The (UHI) effect signifies elevated air and land surface temperatures in urban regions when contrasted with the adjacent rural surroundings [3,4]. It gives rise to issues such as alterations in relative humidity, heightened energy consumption, and thermal discomfort among the human population [5,6]. The intensification of the (UHI) phenomenon originates from human-induced alterations to natural landscapes, leading to subsequent atmospheric and thermophysical adjustments within the urban boundary layer [7]. The emergence of (UHI) phenomenon can predominantly be attributed to the heightened absorption and retention of solar radiation within developed urban structures. Additionally, various contributing factors encompass population density within urbanized zones and the proportion of vegetation present [8].

During Hajj, pilgrims join processions of millions of people from all over the world. Hajj is currently the largest annual gathering in the world. Hajj is one of the five pillars of Islam central to Muslim belief. It is the journey to Mecca that every Muslim must

make at least once in their lifetime if they are able; it is the most spiritual event that a Muslim experiences, observing rituals in the most sacred places in the Islamic world. In the last decade, the capacity of Hajj was around a million and a half. The Kingdom of Saudi Arabia's Vision 2030 aims to boost pilgrim numbers up to 30 million by 2030 [9], approximately 5.4 million annually. The "Doyof Al Rahman Program" is one of the Saudi Vision's realization programs that seeks to enhance the experience and the capacity of Hajj and Umrah, as well as the services provided to pilgrims. However, controlling a crowd of millions performing religious rituals in the same place at the same time is one of Saudi Arabia's key responsibilities.

Walkability is an important concept in sustainable urban design. Improving walkability has health, economic, and environmental benefits [10]. Walkability can also reduce emissions and mitigate climate change. Most of the stages of Hajj occur in a walking environment. Pedestrian conflicts and uncomfortable walking environments are the crucial issues that have been identified, which are due to insufficient pedestrian facilities [11].

Crowd management refers to the practice of controlling the flow of pilgrims from the time they land at the Kingdom's airports until they depart, including moving them between ceremonies and during their Hajj or Umrah journey. One of the main challenges in the management of pedestrians is the ability to determine the carrying capacity of the holy sites, automatically anticipate and identify red areas (overcrowded), and predict areas of suffocation in advance, as well as the possibility of rapid intervention in emergency situations. Many of the solutions being developed today integrate artificial intelligence and real-time data to anticipate crowd dynamics and behaviors during Hajj. The aim of employing artificial intelligence is to improve the basic and additional services provided to the pilgrims, starting with the provision of services for their reception and ending with their farewell through all stages of the journey. The increased temperature precipitates heat-related illnesses and untimely mortality within urban environments [12,13]. Furthermore, the raised air and surface temperatures observed during UHI occurrences lead to heightened average and maximum cooling energy requirements for urban areas. This is a result of the decreased effectiveness of HVAC systems under elevated temperatures, coupled with a notable decrease in the overall level of thermal comfort [14].

Climate change can also impact the walkability of Hajj; rising average temperatures will affect walkability in numerous ways. The increasing frequency and intensity of extreme heat events will impede the ability of residents to walk due to the risk of heat-related illness [11]. Hence, it is necessary to improve the level of services on the aspect of pedestrian facilities. Climate change studies prove that climate change can affect people's physical health in urban areas directly via the increased intensity and frequency of extreme weather events such as heatwaves. Additionally, through worsening air quality, indirectly, it can change the spread of infectious diseases, create threats to food and water, and affect mental health [15].

Climate change is one of the main elements of rapid urbanization [2,6]. Human activity plays a serious role in urban thermal and dynamical environments [12,16]. Urbanization and urban development have significantly impacted the correlation between human existence and the natural environment [17]. The growth of urban populations plays a pivotal role in inducing environmental consequences like climate change, air contamination, and thermal pollution [18].

Addressing climate change involves diminishing the discharge of greenhouse gas emissions responsible for the warming of our Earth [2]. Mitigation approaches encompass renovating structures to enhance their energy efficiency, embracing renewable energy resources like solar, wind, and small-scale hydroelectric power, aiding urban areas in establishing more environmentally sustainable transportation options like bus rapid transit, electric cars, and biofuels and advocating for more ecologically sound land and forest utilization practices. A fundamental strategy for mitigating climate change involves manipulating the Earth's radiation equilibrium by managing both solar and terrestrial radiation.

These methods are referred to as radiative forcing geoengineering technologies, with the primary aim being the stabilization or reduction in temperatures.

Even with numerous endeavors to alleviate the urban heat island (UHI) phenomenon, accurately assessing the efficacy of these mitigation strategies remains challenging due to the limitations of current models during their conception and forecasting phases. This underscores the necessity for the devised tools to comprehensively incorporate a range of intricate events within urban settings while circumventing impractical assumptions and the need for computationally intensive calculations [14].

Recently, Earth observation technology has developed various ways to study climate change [19]. In the majority of preceding research, data from ground-based stations situated in urban locales have been employed to more comprehensively examine shifts in urban climates, including phenomena like the UHI effect [15]. However, for scholars such as the authors of [3,20,21], asserting that remotely sensed measurements yield superior outcomes compared to those derived from interpolating data collected by ground-based stations can be substantiated. In contemporary times, advanced three-dimensional Earth observation systems have been devised to oversee alterations in terrestrial, oceanic, and atmospheric conditions [19,22]. Satellite data's operational range encompasses the electromagnetic spectrum, spanning from visible light and infrared to microwave wavelengths. Earth observation satellites provide scientists with the essential data needed to detect environmental changes on Earth. Because many climate variables can only be measured from space, Earth observation satellites are a vital tool to monitor the effects of climate change on natural ecosystems. Hence, the utilization of multi-system observation networks proves to be a highly efficient technological approach for precisely monitoring shifts in climate.

The precision of remotely sensed climate data with global coverage can always detect the UHIs at the holy sites in Makkah, Saudi Arabia. For this purpose, the present study aims to investigate the urban heat islands in the pedestrian network at the holy sites and understand the factors causing them.

UHI increases the risk of heat-related human illness and mortality [23]. This work investigates the relation between LST and factors such as elevation, vegetation, and built-up indices to assess the impact of UHI on the holy sites. Comparing LST with these factors allows us to create an accurate model to examine the impact of UHI on the pedestrian walkways of the holy sites. This research is an extent of the current knowledge adapting Earth observation (EO) techniques and urban issues for mitigating climate change and pedestrian management.

## 2. Study Area and Methods

Makkah is the holiest city in Islam. It has a population of approximately 4 million [24]; this positions it as the third-most densely populated city in Saudi Arabia, trailing only Riyadh and Jeddah in terms of population [25]. (During the Hajj pilgrimage, this figure is multiplied by over threefold annually.) Makkah is situated within the Hejaz region, at an altitude of 277 m (909 feet) above sea level, positioned at a latitude of 21°23′ north and a longitude of 39°51′ east. It serves as the administrative hub for the Makkah region of Saudi Arabia, and the city is subdivided into 34 districts. Notably, these districts encompass the revered sacred sites, as illustrated in Figure 1. The sacred sites in Makkah are ritually important sites that are mentioned in the Quran and are visited by pilgrims during the annual Hajj. These sites are Mina, Arafat, and Muzdalifah.

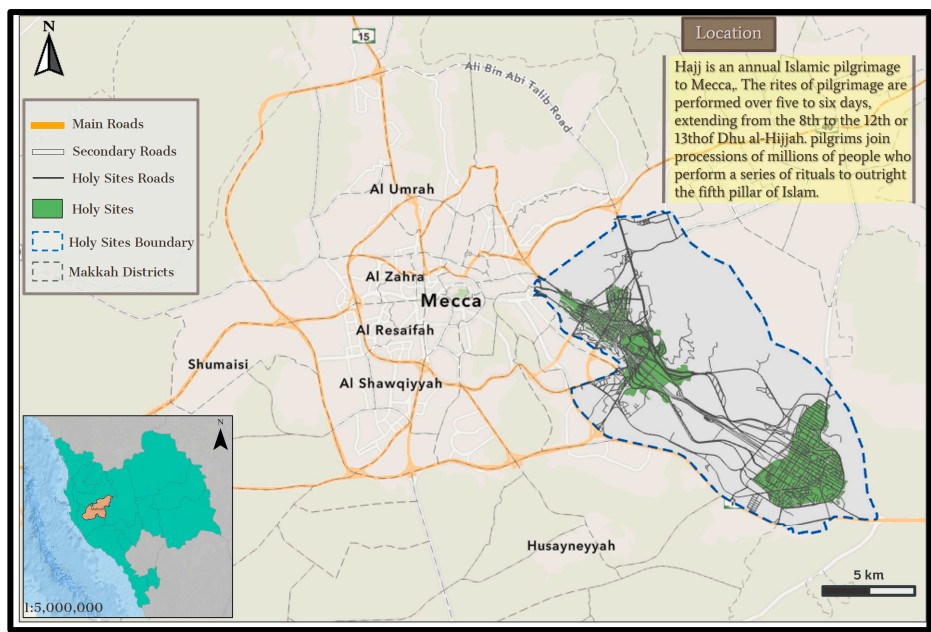

**Figure 1.** Location of the study area.

The particular aims of this research encompass: (1) extracting land surface temperature (LST) data from Landsat 8 OLI and TIRS datasets and generating spatial maps depicting the distribution of LST for the sacred sites in Makkah; (2) generating contour lines of the output LST; (3) identifying the UHI and non-UHI, based on the retrieved LST; (4) mapping the relation between pedestrian networks and LST differences in each holy sites; and (5) integrating the UHIs with the pedestrian network in the holy sites.

Data collected from Landsat 8 OLI and TIRS instruments on 19 August 2019, in Makkah corresponding to the Hajj season prior to the COVID-19 pandemic, were employed to ascertain land surface temperature (LST) and identify urban heat island (UHI) occurrences during the Hajj season. The Landsat 8 OLI and TIRS datasets were developed by the US Geological Survey and were obtained in the format of geographically tagged image files (GeoTIFF). Detailed specifications for the Landsat 8 OLI and TIRS sensors are presented in Table 1. ALOS PALSAR digital elevation model (DEM) data for Makkah were used to obtain the information regarding elevation.

**Table 1.** Landsat 8 OLI and TIRS data specification for the holy sites, Makkah, Saudi Arabia.

| Scene ID | Path/Row | Scene Center Time | Acquisition Date | Sun Azimuth (°) | Cloud Cover (%) |
|---|---|---|---|---|---|
| LC81690452019231LGN00 | 169/45 | 07:43:08.57 | 19 August 2019 | 106.68 | 0% |

The ALOS PALSAR DEM datasets [26] were radiometrically terrain corrected with a resolution of 12.5 m to provide more accurate information about elevation. These elevation data were very helpful in the topographic display of the study area, as well as the representation of the hill shade and the water streamline levels. They were also helpful in understanding the relationships between LST–NDVI and LST–NDBI.

The methodology adopted in this work is based on using satellite imagery bands and undertaking formulations to estimate LST values, then analyzing and comparing the results using correlation analysis and extracting the average temperature values of the pedestrian walk paths to study their initiates. Figure 2 shows the overall methodology of the present study in a flowchart.

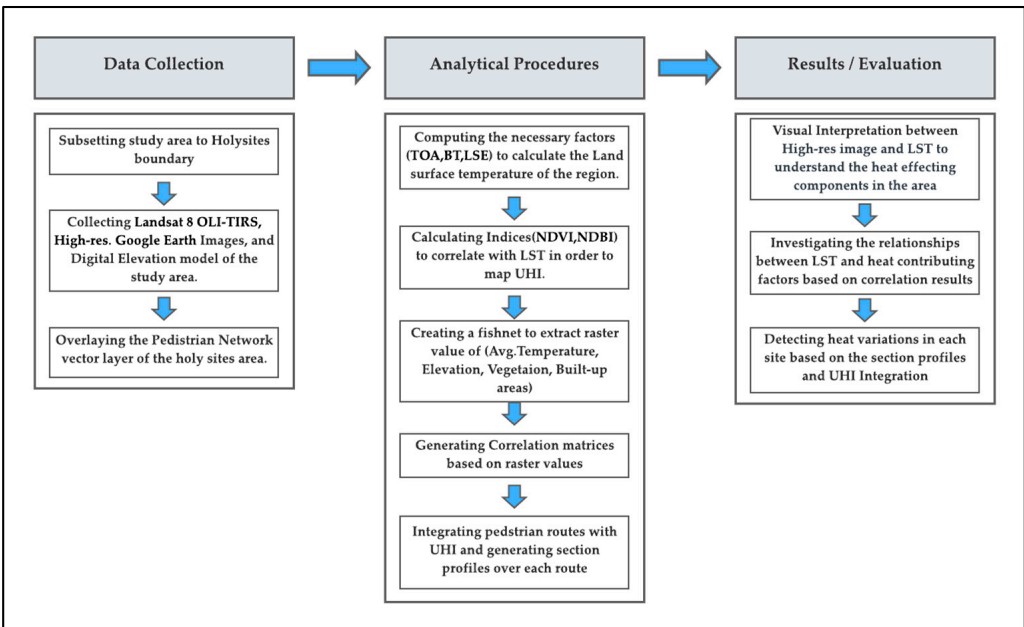

**Figure 2.** The framework of the proposed methodology.

Landsat 8 collection level 1 data were employed for the study. The Landsat 8 TIRS dataset comprises two thermal bands, namely band 10 and band 11. Given the comparatively greater calibration uncertainty linked to band 11, it is advisable for users to focus on utilizing band 10 [27]. The development of NDVI and NDBI involved the utilization of optical bands. For the validation of the NDVI and NDBI indices, reference data were acquired from high-resolution images provided by Google Earth, sourced from the same optical bands. Borders of each holy site's vector layer (provided by the Ministry of Hajj and Umrah) were used as a clipping mask to represent the outputs of each holy site individually. Pedestrian network vector layers were integrated with the LST raster layer of each holy site. Throughout the entirety of the study, ArcGIS Pro 2.9 and QGIS Desktop 3.22 were employed as remote sensing and geographic information system (GIS) software tools for data analysis and to generate the ultimate results.

To reduce the data size within the boundaries of the sacred sites (Arafah, Muzdalifah, and Mina), the satellite data obtained from the Landsat 8 sensor were extracted. First, the raster images of the holy sites were radiometrically corrected to enhance the image quality. The Landsat 8 TIRS images' thermal infrared band, denoted as band 10, possesses a spatial resolution of 100 m. For consistency with the optical bands, this thermal band was subjected to resampling using the nearest neighbor algorithm, resulting in a pixel size of 30 m. To investigate the factors contributing to temperature variations within the study area, the NDVI and NDBI indices were employed to establish their correlation with the derived LST.

LST was retrieved from band 10 of the Landsat 8 OLI and TIRS images of the three holy sites. To calculate LST, five procedures are calculated first. These procedures include estimating the $L\lambda$, $BT$, $Pv$, $\varepsilon$, and $LST$. Following [6,28,29], the calculation procedure was conducted as shown in Table 2. The information of the correlation between LST and other parameters can be found in detail in [30,31].

**Table 2.** Statistic operations to retrieve LST and identify UHI.

| Name | Performance | Definition | |
|---|---|---|---|
| Spectral radiance | $L\lambda$ | $L\lambda = 0.0003342 \times DN + 0.1$ | (1) |
| Brightness temperature | $BT$ | $BT = \left( \frac{K2}{\ln((K1/L\lambda)+1)} \right)$ | (2) |
| Fractional vegetation | $Pv$ | $Pv = \left( \frac{(NDVI-NDVI(min))}{(NDVI(max)-NDVI(min))} \right)^2$ | (3) |
| Effect of geometrical distribution of natural surfaces and internal reflections | $d\varepsilon$ | $d\varepsilon = (1 - \varepsilon s)(1 - Pv)P\varepsilon v$ | (4) |
| Vegetation emissivity | $\varepsilon v$ | $\varepsilon = \varepsilon v Pv + \varepsilon s(1 - Pv) + d\varepsilon$ | (5) |
| Emissivity | $\varepsilon$ | $\varepsilon = 0.004 \times Pv + 0.986$ | (6) |
| Land surface temperature | $LST$ | $LST = \frac{BT}{1+(\lambda\sigma \times BT/(hc))\,\ln\varepsilon}$ | (7) |
| Urban heat islands | *UHI and non-UHI* | $LST > \mu + 0.5 \times \delta$ | (8) |
| Non-urban heat islands | *non-UHI* | $0 < LST < \mu + 0.5 \times \delta$ | (9) |

(1) Lλ represents spectral radiance, measured in units of $\mathrm{Wm^{-2}\,sr^{-1}\,mm^{-1}}$.

(2) BT stands for brightness temperature, measured in Kelvin (K), while Lλ indicates spectral radiance in $\mathrm{Wm^{-2}\,sr^{-1}\,mm^{-1}}$. K2 and K1 denote calibration constants. In the context of Landsat 8 OLI, the values for K1 and K2 are 774.89 and 1321.08, respectively. To convert values from Kelvins to Degrees Celsius, it involves subtracting 273.15 from the "BT in Kelvin" results.

(3) Surface emissivity (ε) was approximated through the employment of the NDVI thresholds approach. The fractional vegetation (Pv) for individual pixels was ascertained based on the NDVI, utilizing the subsequent formula [28]. Here, NDVI min corresponds to the minimum NDVI value (0.2) at which pixels are identified as bare soil, and NDVI max pertains to the maximum NDVI value (0.5) signifying healthy vegetation.

(4) dε accounts for the influence of the geometrical arrangement of natural surfaces and internal reflections. In cases of uneven and diverse surfaces, the value of dε could potentially reach 2%.

(5) εv represents the emissivity of vegetation, εs signifies soil emissivity, Pv stands for fractional vegetation, and P denotes a shape factor with an average value of 0.55 [32].

(6) ε is emissivity; emissivity can be determined using Equations (4) and (5).

Finally, the (7) LST was derived [33]. Here, λ stands for the effective wavelength (10.9 mm for band 10 in Landsat 8 data), σ represents the Boltzmann constant ($1.38 \times 10^{-23}$ J/K), h denotes Planck's constant ($6.626 \times 10^{-34}$ Js), c indicates the velocity of light in a vacuum ($2.998 \times 10^{-8}$ m/s), and ε signifies emissivity.

After calculating the LST, the average temperature of the surface features was determined by creating a fishnet in the data management tools from the arc toolbox used in the ArcGIS environment. This tool generates a grid pattern composed of rectangular cells, with the resulting output available as either polyline or polygon features. By using the boundary of the holy sites' vector layer as a parameter extent to the fishnet, a new point feature class was created with the temperature values extracted from the LST raster output. The process was conducted for each boundary of the holy sites.

UHI and non-UHI were identified using the range of LST determined using Equations (8) and (9) in Table 2 and described in [34]. Here, *μ* represents the mean, and *δ* stands for the standard deviation of LST within the study area.

## 3. Results

### 3.1. Spatial Distribution of LST, NDVI, and NDBI

The distribution of land surface temperature (LST) was categorized into suitable ranges and visually represented using color coding, resulting in the creation of a thermal pattern distribution map illustrating LST across the study area. Table 3 presents the descriptive statistics of LST, NDVI, and NDBI values for the three holy sites. The mean LST values are 38.59 °C for Arafah, 37.31 °C for Muzdalifah, and 35.87 °C for Mina. The threshold value

for UHI generation is 39 °C. The mean NDVI value for the holy sites is 0.06. The mean NDBI value is 0.07.

**Table 3.** Descriptive statistics of LST, NDVI, and NDBI for holy sites.

| Holy Site | LST(°C) | | | NDVI | | | NDBI | | |
|---|---|---|---|---|---|---|---|---|---|
| | Min | Max | SD | Min | Max | SD | Min | Max | SD |
| Mina | 32.7 | 39.5 | 1.29 | −0.01 | 0.25 | 0.01 | −0.51 | 0.29 | 0.05 |
| Muzdalifah | 32.2 | 40.6 | 1.83 | −0.02 | 0.25 | 0.01 | −0.55 | 0.36 | 0.04 |
| Arafah | 35.5 | 42.0 | 1.07 | 0.01 | 0.36 | 0.04 | −0.42 | 0.43 | 0.06 |

*3.2. Correlation Results*

To determine the relationship of the factors affecting the LST in the holy sites, a correlation analysis was established. The NDVI, NDBI, and elevation were three influential characteristics found in the area. Therefore, those three indicators were each compared the values of the LST and then demonstrated in a scatter diagram. The scatter diagram is made of the extracted values' point feature class. A scatter diagram was made for each indicator in comparison to the LST for each holy site.

3.2.1. Mina Correlation Results

In the case of Mina, the correlation was positive and weak in terms of NDVI, as shown in Figure 3. Generally, LST is negatively related to NDVI; however, in this case, LST and NDVI are shown to be positively related. Meanwhile, the correlation was positively strong in terms of NDBI. Generally, LST is positively related to NDBI. On the other hand, the results show that there is a negative correlation between LST and elevation.

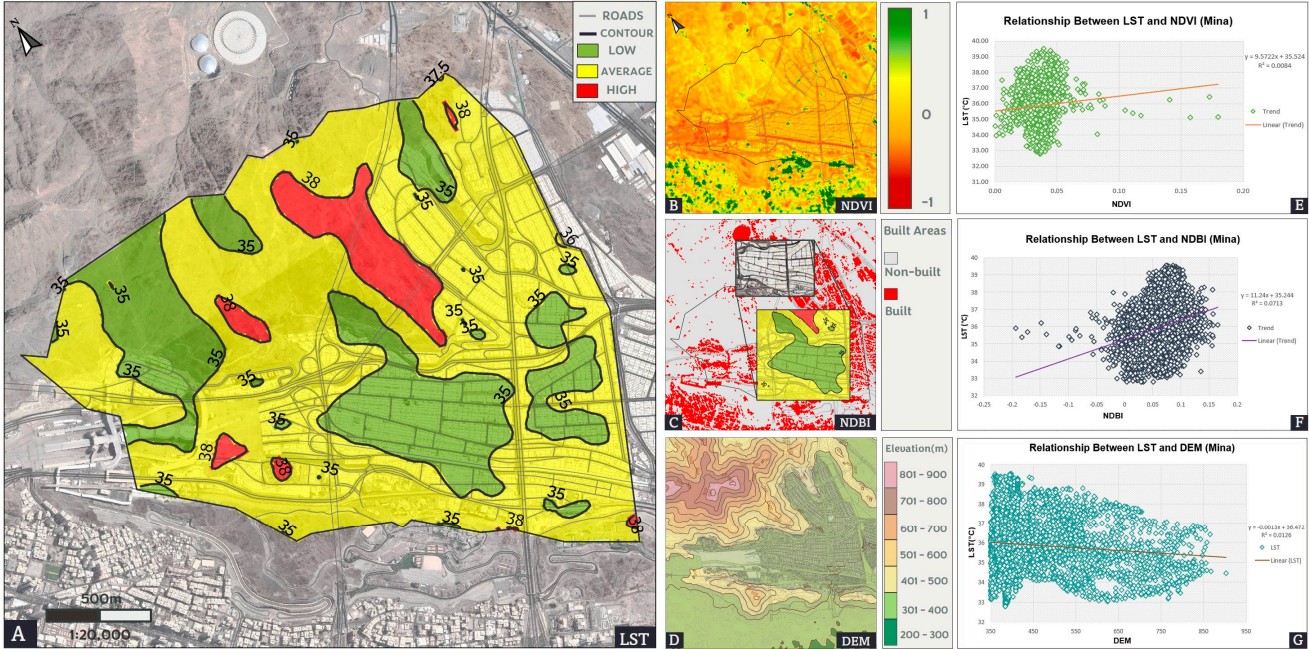

**Figure 3.** The correlation results for Mina: (**A**) LST, (**B**) NDVI, (**C**) NDBI, (**D**) elevation, (**E**) LST and NDVI correlation, (**F**) LST and NDBI correlation, and (**G**) LST and elevation correlation.

3.2.2. Muzdalifah Correlation Results

In the case of Muzdalifah, the correlation between LST and NDVI is positively related, as shown in Figure 4. The strength of this relation is considered weak. In the meantime, the relationship between LST and NDBI is shown to be strongly positive. Concurrently, the results show that there is a negative correlation between LST and elevation.

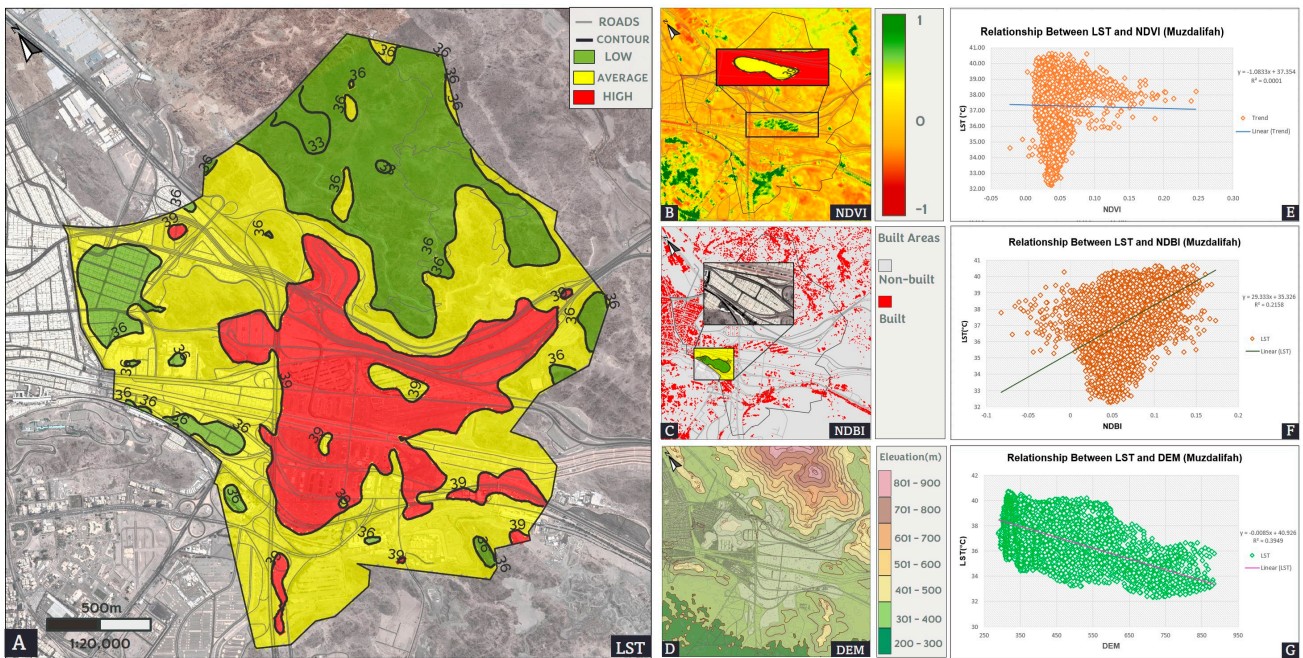

**Figure 4.** The correlation results for Muzdalifah: (**A**) LST, (**B**) NDVI, (**C**) NDBI, (**D**) elevation, (**E**) LST and NDVI correlation, (**F**) LST and NDBI correlation, and (**G**) LST and elevation correlation.

### 3.2.3. Arafah Correlation Results

In the case of Arafah, LST and NDVI are shown to be negatively related in Figure 5. On the contrary, the relationship between LST and NDBI is shown to be strongly positive. Furthermore, there is a negative correlation between LST and NDVI.

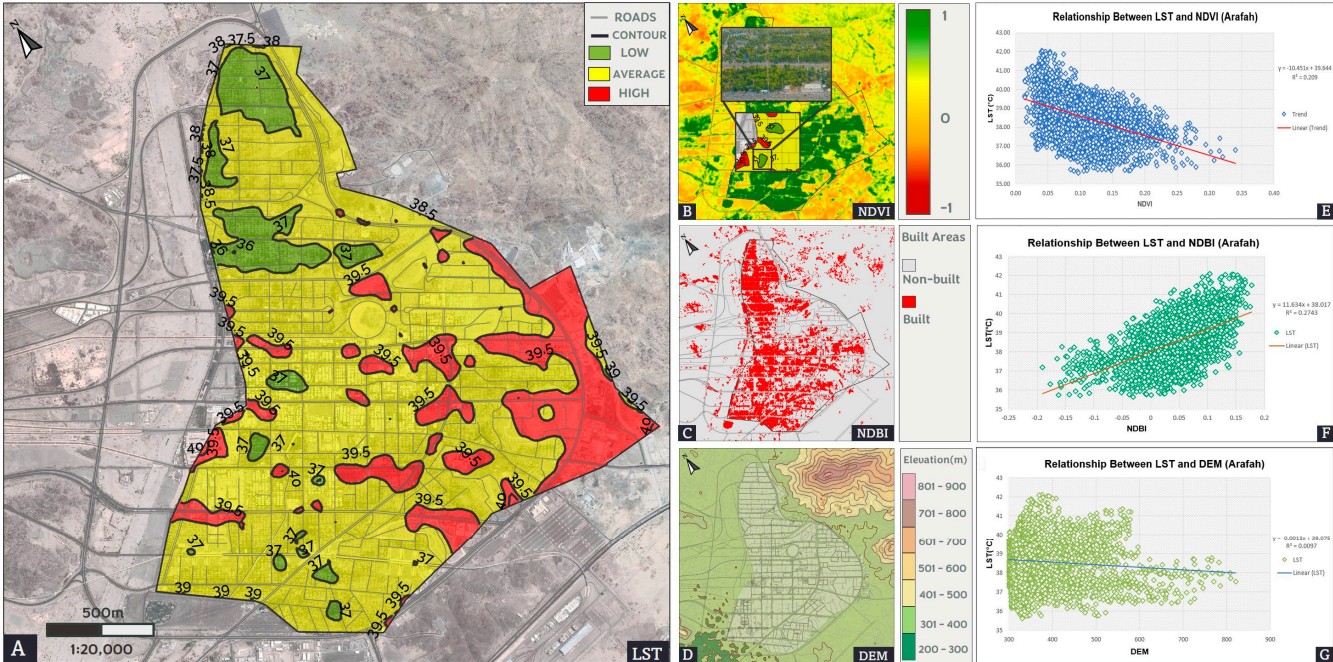

**Figure 5.** The correlation results for Arafah: (**A**) LST, (**B**) NDVI, (**C**) NDBI, (**D**) elevation, (**E**) LST and NDVI correlation, (**F**) LST and NDBI correlation, and (**G**) LST and elevation correlation.

### 3.3. Pedestrian Network Overlay

To detect UHIs over the pedestrian networks in Hajj, the digitized vector line layer was overlayed over the UHI. Using that layer, three section profiles were generated for

each holy site. To represent the urban heat spots on the profile, any temperature value over 39 °C was symbolized as UHI, utilizing the method to map UHIs. Figure 6 shows a map of the pedestrian networks in the holy sites. Figure 7 shows the used section lines for each site to present the LST profile based on the site topography and Hajj journey.

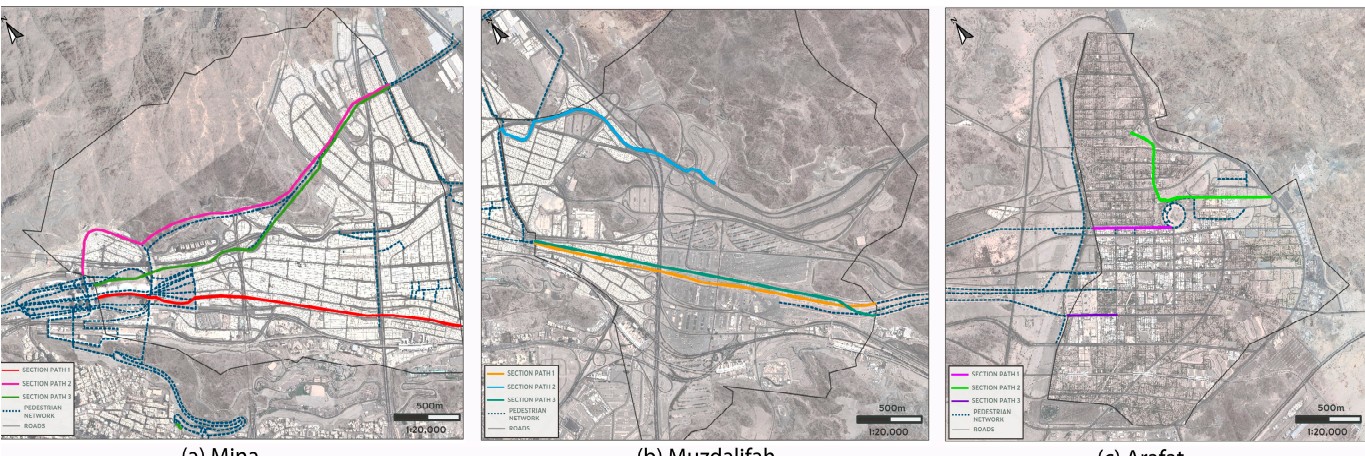

(a) Mina  (b) Muzdalifah  (c) Arafat

**Figure 6.** The selected sections across pedestrian paths to present LST profiles.

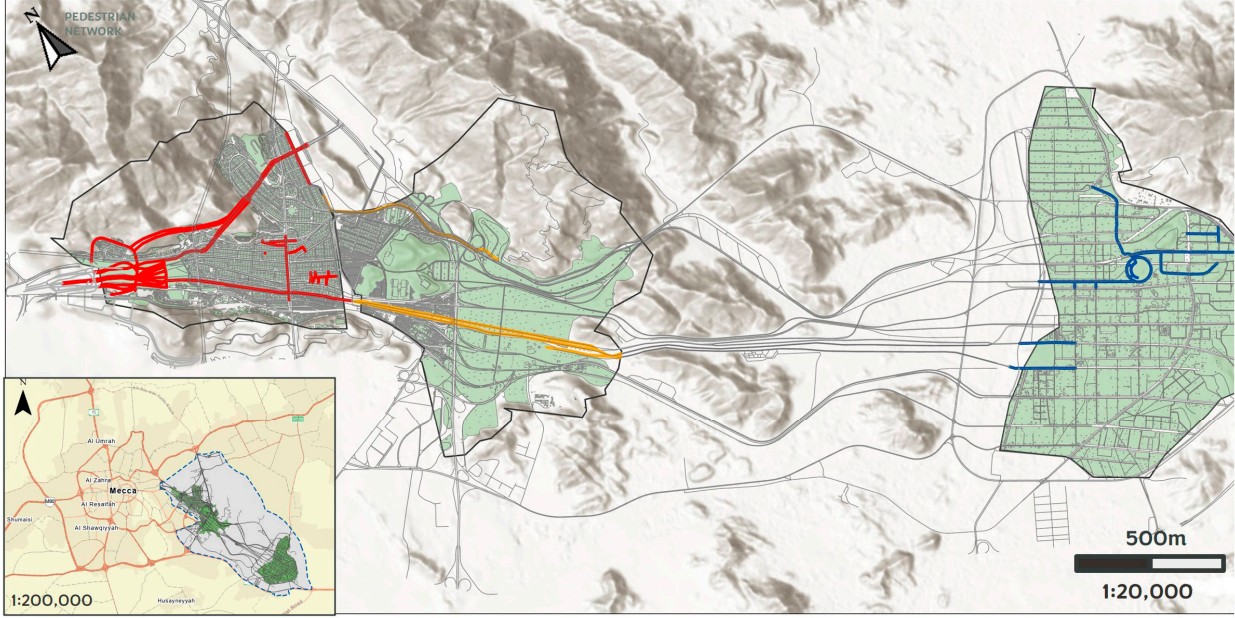

**Figure 7.** Pedestrian network map for three sites. Pedestrians within Mina (red), Muzdalifah (orange), and Arafah (blue).

### 3.3.1. Mina Urban Heat Islands

In the case of Mina, the pedestrian route of Section 1 appears to be free of any heat zones. As for Section 2, the route seems to have an area with an increased surface temperature, as shown in Figure 8. Analogous to the previous section, Section 3 also presents a small area of increased surface temperature.

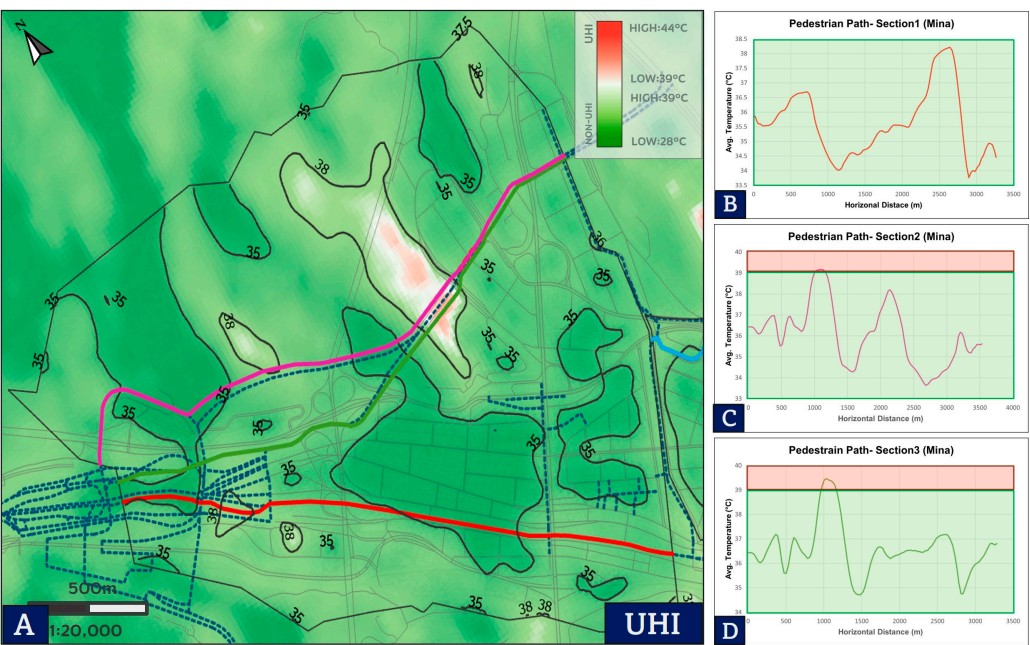

**Figure 8.** (**A**) Mina UHI: (**B**) LST profile for path section 1, (**C**) path section 2, and (**D**) path section 3.

### 3.3.2. Muzdalifah Urban Heat Islands

In the case of Muzdalifah, the pedestrian path in the orange of Section 1 appears to contain larger hotspots, as shown in Figure 9. Section 2 seems to have areas with UHIs and other areas with no UHIs, as shown in the figure. Section 3 is found to be parallel to Section 1; therefore, the heat situation is very similar to the first section, and they face the same dark object issue.

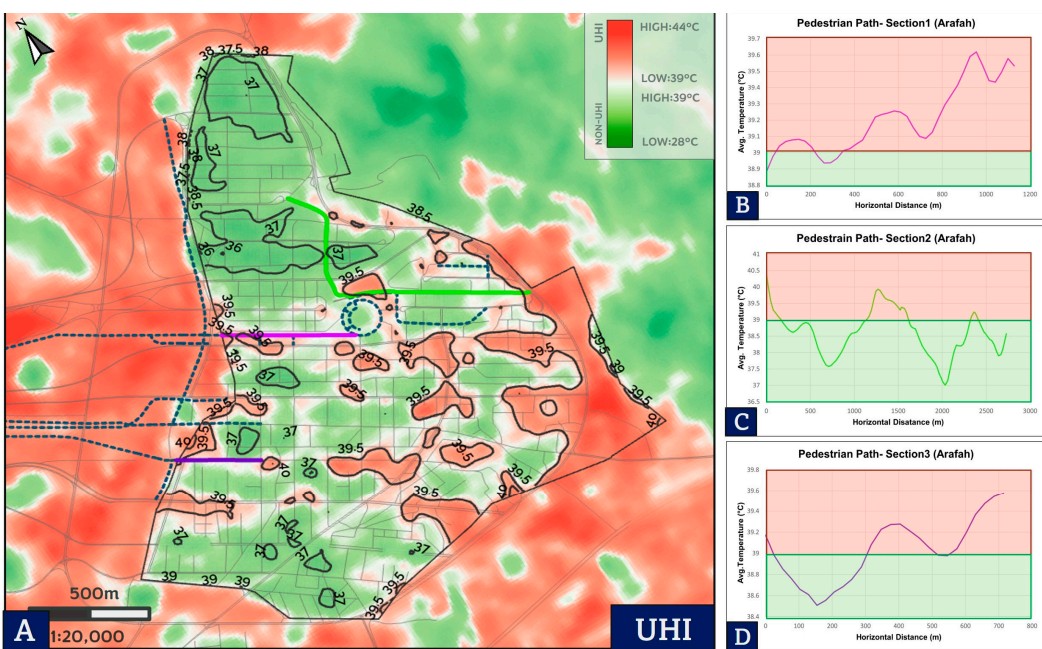

**Figure 9.** (**A**) Muzdalifah UHI. (**B**) UHI profile for Section 1 in orange. (**C**) UHI profile for Section 2 in magenta. (**D**) UHI profile for Section 3 in green.

### 3.3.3. Arafah Urban Heat Islands

In the case of Arafah, the first section seems to contain zones of greatly increased heat, as shown in Figure 10. While Section 2 has areas with UHIs and others with no UHIs,

warmer areas appear to lie between dark surface surroundings. Section 3, in parallel to the previous section, is also found to have heterogeneous characteristics.

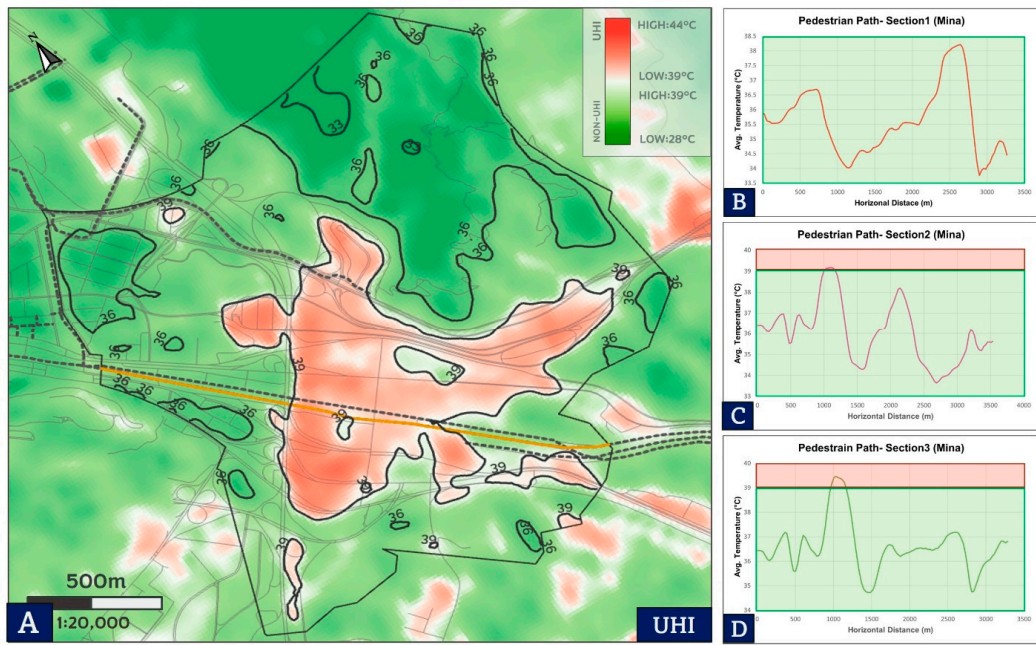

**Figure 10.** (**A**) Arafah UHI. (**B**) UHI profile for Section 1 in orange. (**C**) UHI profile for Section 2 in magenta. (**D**) UHI profile for Section 3 in green.

## 4. Discussion

In this section, the outcomes are examined, and their interpretation is presented within the context of earlier research and the underlying working hypotheses. The implications of the findings are explored within a comprehensive framework. Additionally, potential directions for future research are underscored.

This research investigates the correlation between LST and factors such as NDVI, NDBI, and elevation to provide a sustainable solution and help decision makers improvise strategies that will ease the management of the flow of Hajj. Landsat 8 OLI and TIRS data were used to investigate the UHI on the pedestrian network at the holy sites in Makkah, Saudi Arabia and to interpret the relationship of the factors causing the UHI effect. The result of this research provides a measurable indicator of the impact of climate change on a small scale at gathering events in specific conditions, as shown in [22,35]. Generally, the results of the spatial distribution of LST, NDVI, and NDBI from Table 2 reveal little heterogeneity in LST due to the use of different materials in each holy site. The most widely known and frequently used land cover metric is NDVI, which calculates the percentage of vegetation in a pixel [36,37]. Ref. [4] argued that in desert environments, the UHI can deviate from the standard distribution, where it was previously noted from global assessments that more populated areas have relatively higher LST compared to the adjacent countryside [38]. Generally, all three holy sites have a similar range of statistical data regarding LST, NDVI, NDBI, and elevation. UHI zones were identified through LST, which appeared enlarged in areas within Arafat, while they developed in the central region of Muzdalifah; on the other hand, they appeared in very small areas in Mina. The use of dark surfaces is mostly responsible for LST generation. The presence of vegetation and the use of white roofs in built-up areas reduces the LST level. Therefore, it makes sense that expanding the city's green spaces will aid in UHI mitigation [2].

According to the results of Mina, the correlation of LST and NDVI is positive and weak. The strength of this relation is considered weak due to the lack of vegetation in Mina. In general, the region has a great amount of vegetation cover due to climate, soil type, water availability, and encroachment on vegetation cover regarding appearance.

The variation in this relationship could be affected by the spatial resolution, latitudinal extension, or seasonal variation. For example, the Hajj season depends on the moon calendar, which affects the beginning of the season each year. [39] discusses the impact of spatial resolution and temporal resolution on UHI analysis. Although multiple data fusion techniques were applied using different earth observation datasets to estimate LST, Landsat imagery remains the most used dataset due to data availability and as open source data. However, the usefulness of Landsat spatial resolution with 30 m and other spatial resolution datasets up to 50 m were evaluated and recommended by [39]. Furthermore, the terrain factors affect the LST as approved in this study, and the degree of influence needs further investigation as discussed in [40].

Meanwhile, NDBI has a positive correlation with LST. The temperature values seem lower in built-up areas than in other bare lands. This is due to the white materials used in the tent roofs in Mina, which absorb less heat. On the other hand, LST and elevation were shown to have a negative correlation, which is consistent with the results of another existing study [23]. Mina has the most variations in elevation among the holy sites. However, there is a slight increase in surface temperature in elevated areas. The increase might be caused by artificial environmental disturbances, such as built-up lands.

In regard to detecting UHI pedestrian paths, the pedestrian route of Section 1 appears to be free of any heat zones, as shown in Figure 11. The white materials used to play an important role in keeping this route a non-UHI zone, as discussed in [29]. As for Section 2, the route seems to have an area with an increased surface temperature, as shown in Figure 11. The shown warmer area is within the tunnel area of Mina. The increased temperature could be caused by artificial environmental disturbances. Analogous to the previous section, Section 3 is also found to have a small area of increased surface temperature.

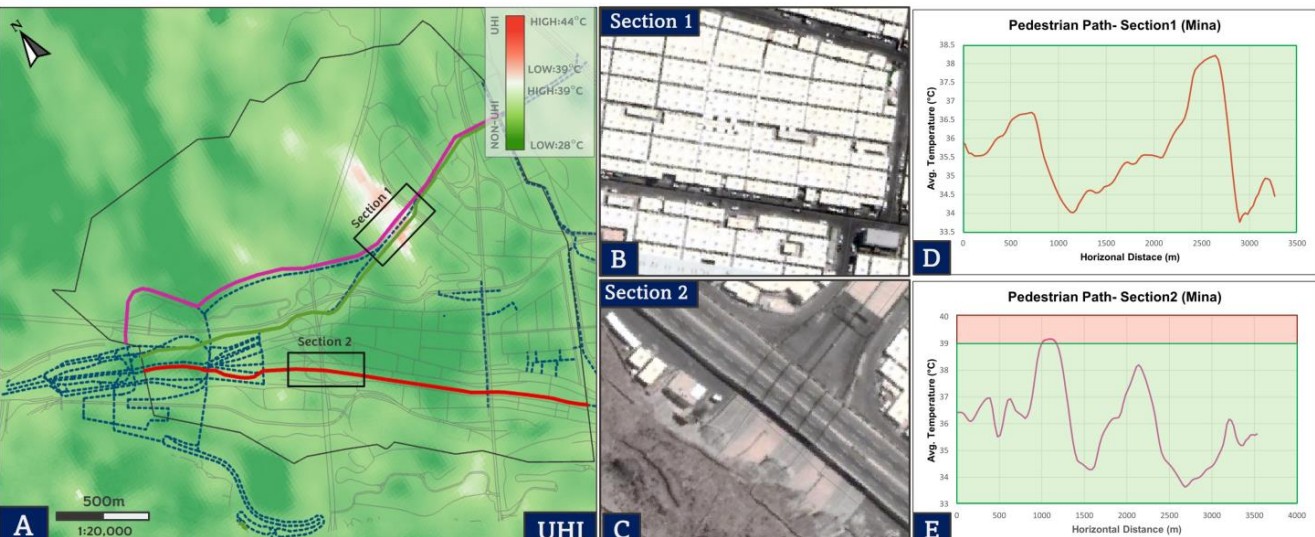

**Figure 11.** Causes of UHI in Mina: (**A**) Mina UHI; (**B**) use of white materials in tents; (**C**) use of concrete [41]; (**D**) UHI profile for Section 1, in orange; and (**E**) UHI profile for Section 2, in magenta.

According to the results of Muzdalifah, the correlation between LST and NDVI is positively related. The strength of this relation is considered weak due to the lack of vegetation in Muzdalifah. New initiatives such as Green Holy Sites began to afforest the walking paths of Muzdalifah in March 2022. The afforestation process recently began to make an impact in decreasing the temperature, as shown in Figure 12. Therefore, the improvement in LST took the form of a temperature decrease when just vegetation cover was introduced to the area.

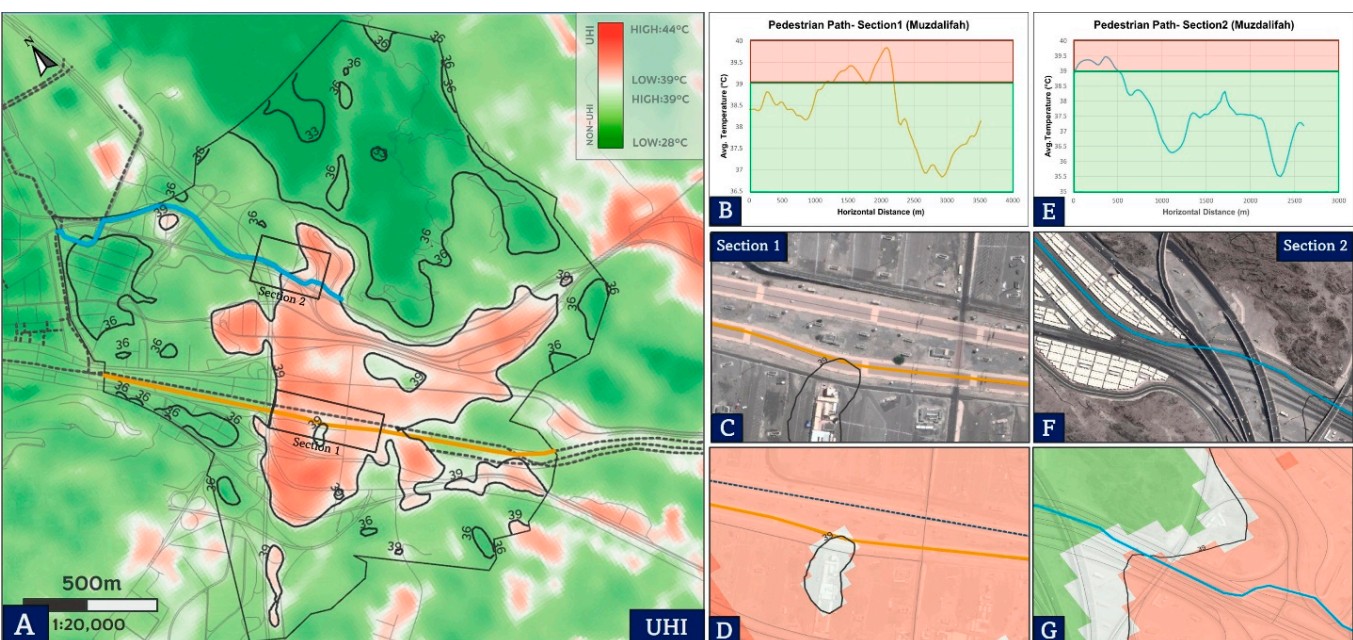

**Figure 12.** Causes of UHI in Muzdalifah: (**A**) Muzdalifah UHI, (**B**) path Section 1, (**C**) true-color image of Section 1, (**D**) image of Section 1 with UHI overlay, (**E**) path Section 2, (**F**) true-color image of Section 2, and (**G**) image of Section 2 with UHI overlay.

In the meantime, the relationship between LST and NDBI was shown to be strongly positive. Muzdalifah has the lowest number of bult-up areas compared to the other holy sites. The use of white materials in buildings has also proven to be effective in absorbing heat in Muzdalifah. Other built-up areas use dark materials that absorb more heat, and they seem to raise the surface temperature steadily in Muzdalifah.

Concurrently, the results show that LST and elevation are negatively correlated. The impact of elevation on (LST) primarily stems from air temperature; as elevation rises, temperature experiences a gradual decline. There is an obvious trend: LST decreases gradually with increasing elevation. Regarding detecting UHI pedestrian paths, the pedestrian path in orange in Section 1 appears to contain larger hotspots, as shown in Figure 12. Dark surfaces such as asphalt can lead to a high absorption of heat, as discussed in [1].

This heat issue can be solved by adding more green cover to reduce heat islands, i.e., the Green Holy Sites Initiative, as one of the main climate mitigation solutions, as discussed in [42]. However, the increase in vegetation extent is not an option due to limited spaces and the Hajj program. In addition, other factors should be included in the solutions, such as energy type and building code, to ensure sustainable development [1,43,44]. Section 2 seems to have areas with UHIs and other areas with no UHIs, as shown in the figure. Warmer areas appear to lie between dark surface surroundings. The non-UHI parts of the route lie within the light-roofed tents, which help cool the environment. Section 3 is found to be parallel to Section 1; therefore, the heat situation is very similar to that of the first section, where the same dark object issue is faced.

According to the results of Arafah, LST and NDVI are negatively related. The strength of this relation is considered strong due to the large areas of vegetation in Arafah. Arafah is considered to have the highest extent of vegetation of the holy sites, as shown in Figure 13. These large areas of vegetation play an important role in the reduction in surface temperature, as proven in [45].

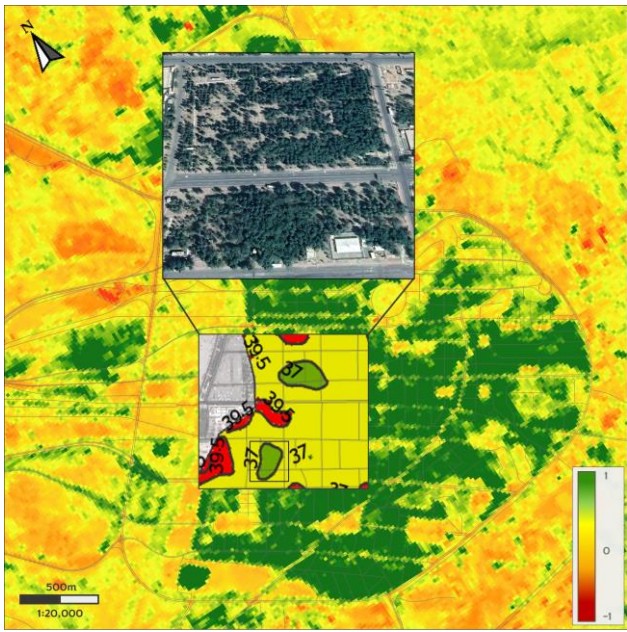

**Figure 13.** Vegetation cover in Arafah introduced in the area as environment mitigation.

On the contrary, the relationship between LST and NDBI was shown to be strongly positive. Arafah has many different types of built-up areas. The noticeable use of dark concrete surfaces in some areas has led to a sharp increase in surface temperature. These surfaces lead to a higher absorption in temperature and cause the formation of heat spots in the area, as shown in Figure 14. Furthermore, there is a negative correlation between the LST and elevation. The holy site of Arafah has a lower elevation compared to Muzdalifah and Mina, as shown in Figure 15. Smaller topographic fluctuations in the Hajj area mean a greater amount of human activities.

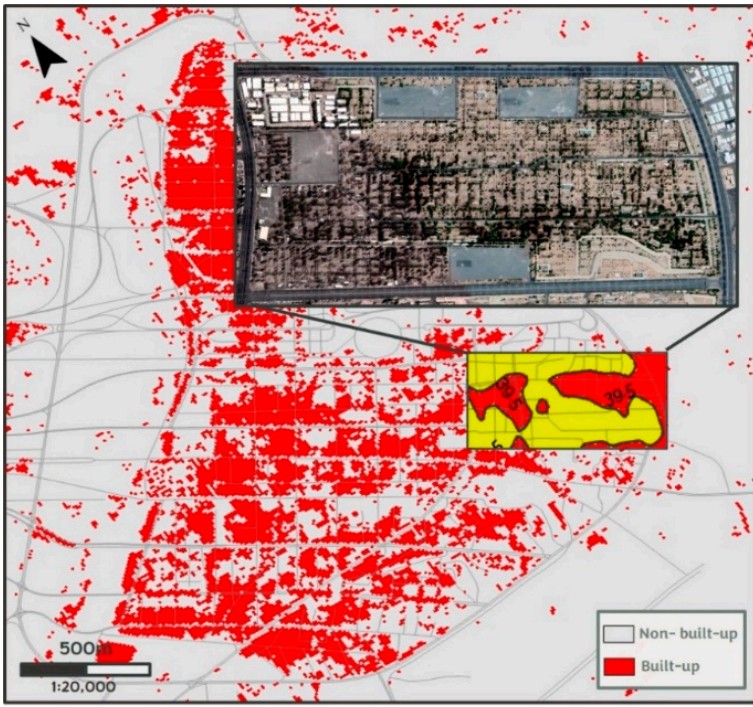

**Figure 14.** Built-up areas in Arafah showing the impact of building materials on UHI in the area.

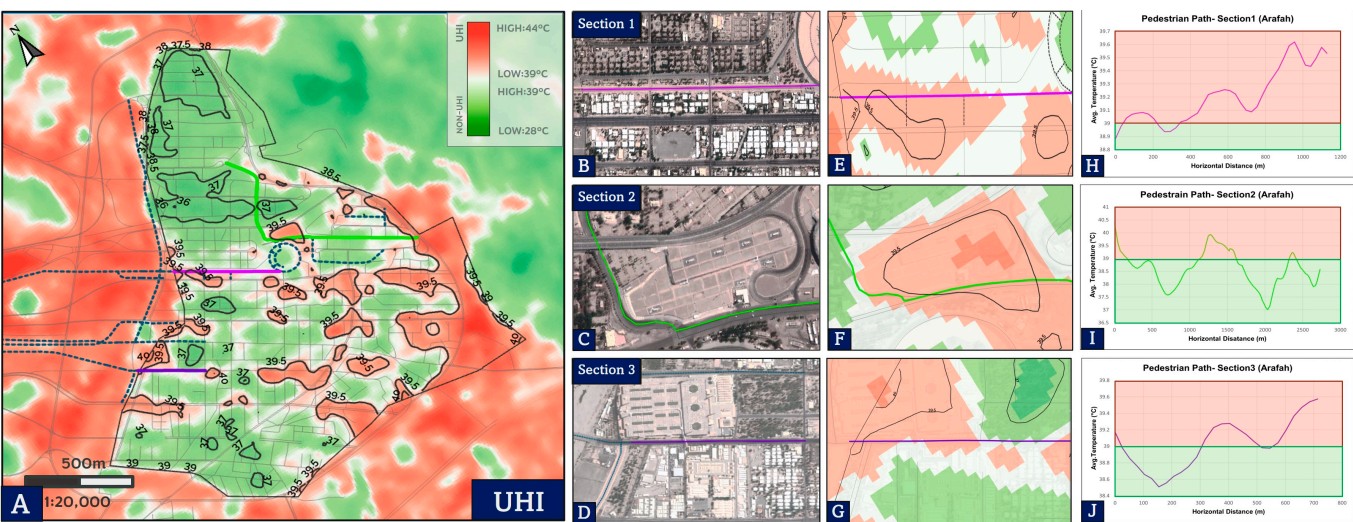

**Figure 15.** Causes of UHI in Arafah: (**A**) Arafah UHI, (**B**) true-color image of Section 1, (**C**) true-color image of Section 2, (**D**) true-color image of Section 2, (**E**) path Section 2, (**F**) image of Section 2 with UHI overlay, (**G**) image of Section 2 with UHI overlay, (**H**) path Section 1, (**I**) path Section 2, and (**J**) path Section 3.

In the case of Arafah, the first section seems to contain a large amount of heat zones. Dark surfaces are profusely found in this route. While Section 2 has areas with UHIs and others with no UHIs, warmer areas appear to lie between dark surface surroundings. The non-UHI parts of the route are within a vegetated green area, which helps cool the environment, as shown in Figure 15. Section 3, in parallel to the previous section, was also found to have heterogeneous characteristics. This route is adjacent to the Al-Namera Mosque area, whose excess heat is released by air conditioning units, worsening the urban heat island effect. On the other hand, this route is also located within areas with high vegetation, which reduces the heat zone by a little.

For Arafah and Muzdalifah, LST shows a negative correlation with NDVI. Nonetheless, the correlation is positive and weak in Mina's case. This may be due to the lack of vegetation in Mina. The correlation between LST and NDBI is found to be strongly positive in the case of Mina, Muzdalifah, and Arafah. This is due to the use of dark materials and surfaces that absorb more heat, which causes the LST to rise steadily. The correlation between LST and elevation was found to be negative in the case of Mina, Muzdalifah, and Arafah.

The studied sections in Mina were found to contain the least heat spots. This may be due to the use of white materials in built-up areas, such as tents. Meanwhile, in Muzdalifah, the sections were found to contain larger heat spots than Mina. This may be due to the use of darker surfaces, such as asphalt, which can lead to a high absorption of heat. In the case of Arafah, the sections were found to contain the largest amount of heat spots, which is due to the dark surfaces that were profusely found in the area.

## 5. Conclusions

In this work, Earth observation technology was used to investigate the impact of land surface temperature in holy sites. Earth observation techniques prove their usefulness in mapping UHI on a pedestrian level, as shown in all cases. The influence of factors such as vegetation, used materials, and elevation on LST was discussed. There is a certain correlation between LST, NDVI, NDBI, and elevation in the study area. These relationships were interpreted quantitively using a scatter diagram. A clear pattern is evident: LST exhibits a gradual reduction as elevation increases. The correlation between elevation and LST is primarily driven by air temperature, with temperature gradually diminishing as elevation rises. This observed trend can be attributed to the fact that the upper layers of the tropospheric atmosphere, situated farther from the ground, absorb less long-wave

radiation energy. Consequently, due to the decreased heat retention by the atmosphere, lower temperatures are experienced. In addition, UHIs found within the pedestrian paths were investigated by modeling the change in surface temperature along the path in a graph. UHIs were overlayed with a pedestrian path layer, and three sections were modeled from each site. To represent the urban heat spots on the profile, any temperature value over 39 °C was symbolized as a UHI by utilizing the method to map UHIs.

In future, there is scope for conducting numerous supplementary research endeavors. To begin with, there is potential for exploring alternative methodologies or varying spatial resolutions for the retrieval of LST. Secondly, the in situ LST data could be collected during the same satellite overpass, serving as a means of calibration and validation for the estimation of LST. Thirdly, the correlation could be analyzed with the inclusion of more parameters. Finally, TIRS as a thermal band could be installed on airborne devices to measure areas on a bigger scale. Through this approach, mapping framework can aid urban planning experts in making informed decisions about the adapted strategies. This selection considers the projected local climate conditions. Using Earth observation technologies is essential for studying climate change. The performance of different UHIs greatly depends on the urban fabric, the local climatic conditions, even the time of the day, and human activities. Management of the energy type and conception is needed.

**Author Contributions:** Conceptualization, A.F. and A.I.; methodology, A.F. and A.I.; software, A.F.; validation, A.F. and A.I.; formal analysis, A.F. and A.I.; investigation, A.F. and A.I.; resources, A.F and A.I.; data curation, A.F.; writing—original draft preparation, A.F. and A.I.; writing—review and editing, A.F. and A.I.; visualization, A.F. and A.I.; supervision, A.F.; project administration, A.F.; funding acquisition, A.F. All authors have read and agreed to the published version of the manuscript.

**Funding:** The Deanship of Scientific Research (DSR) at King Abdulaziz University (KAU), Jeddah, Saudi Arabia has funded this project, under grant number HO: 20-137-1443.

**Institutional Review Board Statement:** Not applicable.

**Informed Consent Statement:** Not applicable.

**Data Availability Statement:** The data that support the findings of this study are available on request from the corresponding author, [AF]. However, The used satellite images were downloaded from USGS: https://earthexplorer.usgs.gov, accessed on 11 July 2023.

**Acknowledgments:** This research was funded by the Deanship of Scientific Research at King Abdulaziz University. This research was supplied by The Ministry of Hajj and Umrah for the land use data at the holy sites. The author is also indebted to Kidana for providing valuable knowledge about the nature of Hajj.

**Conflicts of Interest:** The authors declare no conflict of interest.

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
