# Peer review of "Detecting Land Surface Temperature Variations Using Earth Observation at the Holy Sites in Makkah, Saudi Arabia"

_sustainability, doi:10.3390/su151813355_

Round 1

Reviewer 1 Report

The author produces a detailed study of the three holy sites for the hajj pilgrims. The results are interesting and satisfying. Some minor issues notes are as follows:

·         Check about the copyright issues if you used some figure or photo from outside (Like Figures 1 and 2)

·         Title improvement. Include the land surface temperature or thermal characteristics or some other for your LST result along with the UHI study.

·         Improve flow chart figure to look much better.

·         Why does LST show a weak or some positive correlation in Mina and Muzdalifah

·         Figure 5: legend not readable.

·         Copyright issue of photos in Figures 11 to 16.

·         Figures are not placed serially.

Minor improvement. Check grammatical corrections carefully.

Author Response

Response to reviewer 1 comments

General comments

Thank you for your comments and engagements. The manuscript has been updated as per the reviewers recommend where possible. We provide a justification for each question highlighting the modifications made in red in the manuscript. We have responded to each criterion separately and modified and corrected the thesis accordingly. A comprehensive editing for the whole manuscript has been completed to increase readability and clarity Using the MDPI English service. Kindly, find the updated references in the manuscript and some references used for justification in our responses at the end of the response template.

Open Review

Quality of English Language

( ) I am not qualified to assess the quality of English in this paper
( ) English very difficult to understand/incomprehensible
( ) Extensive editing of English language required
( ) Moderate editing of English language required
(x) Minor editing of English language required
( ) English language fine. No issues detected

Comments and Suggestions for Authors

The author produces a detailed study of the three holy sites for the hajj pilgrims. The results are interesting and satisfying. Some minor issues notes are as follows:

  • Check about the copyright issues if you used some figure or photo from outside (Like Figures 1 and 2)

Answer:

Thank you for your comments. The copy right issue in figure 1 and figure 2 has been resolved. The caption was modified adding the source. Now, it reads as follows: 

“Figure 1. The holy sites during Hajj activities (Source [1]). ”

And

“Figure 2. The timeline journey of Hajj (Source: [2])..”

  • Title improvement. Include the land surface temperature or thermal characteristics or some other for your LST result along with the UHI study.

Answer:

We agree with you. The title was updated to reflect the actual implementation of LST and used data. Now, it reads as follows:   
“Detecting Land Surface Temperature Variations using Earth Observation in the Holy Sites of Makkah, Saudi Arabia”

  • Improve flow chart figure to look much better.

Answer:

Thank you for your comments. The flowchart (Figure 4) was updated to increase the readability and visibility. 

  • Why does LST show a weak or some positive correlation in Mina and Muzdalifah

Answer:

Thank you for your comments. The answer to LST correlation can be found in the discussion section in three locations as mentioned below:

In line 332: “According to the results of Mina, the correlation of LST and NDVI was positive and weak. The strength of this relation is considered weak due to the lack of vegetation in Mina.”

In line 354 “According to the results of Muzdalifah, the correlation between LST and NDVI is positively related. The strength of this relation is considered weak due to the lack of vegetation in Muzdalifah.”

In line 383 ‘According to the results of Arafah, LST and NDVI are negatively related. The strength of this relation is considered strong due to the large areas of vegetation in Arafah. Arafah is considered to have the highest extent of vegetation of the holy sites, as shown in Figure 16. These large areas of vegetation play an important role in the reduction in surface temperature, as proven in [3].”

  • Figure 5: legend not readable.

Answer:

Thank you for your comments. The legend in Figure 5 has been elucidated to be readable.

  • Copyright issue of photos in Figures 11 to 16.

Answer:

Thank you for your comments. The copy right issue in figure 11 to 16 has been solved. The copyright is mow mentioned in the figure’s caption.

  • Figures are not placed serially.

Answer:

Thank you for your comments. The order of figures was checked for consistency and clarity. Now, the figures are numbered serially.

Reviewer 2 Report

The authors need to thoroughly review the article and improve as per the standard of the journal. 

The figure quality needs to improve. I have marked the correction on the manuscript which need to addresses.

The results and discussion need to improve and conclusions need to write specifically based on the outcome of the study.  The remedial measure also needs to suggest for the UHI areas. 

What comfort criteria are considered in this study need to be mentioned?

The authors need to thoroughly review the article and improve as per the standard of the journal.  

Author Response

Response to reviewer 2 comments

General comments

Thank you for your comments and engagements. The manuscript has been updated as per the reviewers' recommendation where possible. We provide a justification for each question highlighting the modifications made in red in the manuscript. We have responded to each criterion separately and modified and corrected the thesis accordingly. Comprehensive editing for the whole manuscript has been completed to increase readability and clarity Using the MDPI English service. Kindly, find the updated references in the manuscript and some references used for justification in our responses at the end of the response template.

Open Review

Quality of English Language

( ) I am not qualified to assess the quality of English in this paper
( ) English very difficult to understand/incomprehensible
(x) Extensive editing of English language required
( ) Moderate editing of English language required
( ) Minor editing of English language required
( ) English language fine. No issues detected

Yes

Can be improved

Must be improved

Not applicable

Is the content succinctly described and contextualized with respect to previous and present theoretical background and empirical research (if applicable) on the topic?

( )

( )

(x)

( )

Are all the cited references relevant to the research?

( )

( )

(x)

( )

Are the research design, questions, hypotheses and methods clearly stated?

( )

( )

(x)

( )

Are the arguments and discussion of findings coherent, balanced and compelling?

( )

( )

(x)

( )

For empirical research, are the results clearly presented?

( )

( )

(x)

( )

Is the article adequately referenced?

( )

( )

(x)

( )

Are the conclusions thoroughly supported by the results presented in the article or referenced in secondary literature?

( )

( )

(x)

( )

Comments and Suggestions for Authors

The authors need to thoroughly review the article and improve as per the standard of the journal. 

Answer:

Instructions for authors were checked and reviewed carefully to ensure the template alignment with the MDPI stander.

The figure quality needs to improve. I have marked the correction on the manuscript which need to addresses.

Answer:

Thank you for your comments. Image quality was improved, and additional information was added as suggested to increase the readability and remove ambiguity.

The results and discussion need to improve and conclusions need to write specifically based on the outcome of the study.  The remedial measure also needs to suggest for the UHI areas. 

Answer:

Thank you for your comments. The manuscript was updated and supported adding references in the above per as the reviewer's suggestion. Additionally, the manuscript was supported with extra information and references to support the state of the art of UHI and LST analysis. Therefore, the conclusion was modified to represent the importance of the framework, the significance of the finding and future work.

What comfort criteria are considered in this study need to be mentioned?

Answer:

Thank you for your comments. This research seeks to evaluate the UHI phenomenon in the Middle Eastern environment and urban context. Using the EO approach allows for the incorporation of geospatial and other ancillary data in addition to the base VHR imagery. UHI increases the risk of heat-related human illness and mortality. This work investigates the relation between LST and parameters such as elevation, vegetation extent and built-up, to assess the impact of UHI on the unique and challenging environment. Following many researchers [4-6], this research evaluates the change of LST in significant geographic locations to help local authorities provide a better practice of environmental mitigation. However, the term comfort has been deleted to remove ambiguity and more consistency.

Comments on the Quality of English Language

The authors need to thoroughly review the article and improve as per the standard of the journal.  

Answer:

Instructions for authors were checked and reviewed carefully to ensure the template alignment with the MDPI stander. To enhance the readability and increase clarity, the whole manuscript has been edited by a native English editor using the MDPI English service.

Submission Date

11 July 2023

Date of this review

23 Jul 2023 15:08:29

Reviewer 3 Report

Review: Detecting Urban Heat Islands on Urban pedestrian network using Earth observation technology in the holy sites in Makkah, Saudi Arabia

This reviewer thanks the authors of the above paper for an intriguing study of the landscape during an important religious pilgrimage. The attention to sustainable practices and human health is clearly essential.

This reviewer's primary issue with the paper is the documentation of correlation analysis throughout. The process for obtaining a correlation coefficient (and the reviewer will assume Pearson’s Correlation Coefficient) is not specified in the methods section. Nor has a test for Gaussian properties been undertaken per variate – Table 3 does not show average values; thus, readers cannot determine Normal distribution properties themselves. Further, the authors do not publish the coefficient results per covariate – nor the linear correlation equations. The figures indicate that standard outlier analysis has not been conducted – and that some of the bivariate plots do not exhibit linear correlation at all – some non-linear characteristics are clearly visible. This reviewer recommends that a statistician assist the authors in preparing their visualisations and documenting their results.

This reviewer is unsure that the author’s interpretation of the statistics is valid.

Upon that work, the Figures will need rework regarding maps legends and choropleths. For the LST maps, what have the authors defined as low, medium and high correlation (note: some standard definitions exist)? For NDVI, green is more intuitive for high NDVI values (i.e., healthy vegetation). Many Figure map legends must provide the map feature explanation that cartographic practice requires.

In terms of minor adjustments, this reviewer requests the following:

-        The methods section needs reorganising to clarify the methods used – the summary of lines 123 to 128 does not cover later method explanation.

-        A fuller explanation for why a correlation analysis of NDVI (etc) with LST is essential for the study. What is the physical basis for that relationship – and will it help indicate in the forthcoming results? The use of NDBI similarly needs much more explanation.

-        A discussion on Landsat spatial resolution – and the inherent limitations would be helpful.

-        The Figure numbers in the text do not always match the Figure captions.

Noting these difficulties, the reviewer did not review the contents of the Discussion sections.

Minor grammatical errors to be corrected - for example, line 15 - change 'increase the number pilgrims' to 'increase the number of pilgrims'.

Author Response

Response to reviewer 3 comments

General comments

Thank you for your comments and engagements. The manuscript has been updated as per the reviewers' recommendation where possible. We justify each question highlighting the modifications made in red in the manuscript. We have responded to each criterion separately and modified and corrected the thesis accordingly. Comprehensive editing for the whole manuscript has been completed to increase readability and clarity Using the MDPI English service. Kindly, find the updated references in the manuscript and some references used for justification in our responses at the end of the response template.

Open Review

Quality of English Language

( ) I am not qualified to assess the quality of English in this paper
( ) English very difficult to understand/incomprehensible
( ) Extensive editing of English language required
( ) Moderate editing of English language required
(x) Minor editing of English language required
( ) English language fine. No issues detected

Yes

Can be improved

Must be improved

Not applicable

Is the content succinctly described and contextualized with respect to previous and present theoretical background and empirical research (if applicable) on the topic?

( )

( )

(x)

( )

Are all the cited references relevant to the research?

( )

(x)

( )

( )

Are the research design, questions, hypotheses and methods clearly stated?

( )

( )

(x)

( )

Are the arguments and discussion of findings coherent, balanced and compelling?

( )

( )

(x)

( )

For empirical research, are the results clearly presented?

( )

( )

(x)

( )

Is the article adequately referenced?

( )

(x)

( )

( )

Are the conclusions thoroughly supported by the results presented in the article or referenced in secondary literature?

( )

( )

(x)

( )

Comments and Suggestions for Authors

Review: Detecting Urban Heat Islands on Urban pedestrian network using Earth observation technology in the holy sites in Makkah, Saudi Arabia

This reviewer thanks the authors of the above paper for an intriguing study of the landscape during an important religious pilgrimage. The attention to sustainable practices and human health is clearly essential.

This reviewer's primary issue with the paper is the documentation of correlation analysis throughout. The process for obtaining a correlation coefficient (and the reviewer will assume Pearson’s Correlation Coefficient) is not specified in the methods section. Nor has a test for Gaussian properties been undertaken per variate – Table 3 does not show average values; thus, readers cannot determine Normal distribution properties themselves. Further, the authors do not publish the coefficient results per covariate – nor the linear correlation equations. The figures indicate that standard outlier analysis has not been conducted – and that some of the bivariate plots do not exhibit linear correlation at all – some non-linear characteristics are clearly visible. This reviewer recommends that a statistician assist the authors in preparing their visualisations and documenting their results.

This reviewer is unsure that the author’s interpretation of the statistics is valid.

Upon that work, the Figures will need rework regarding maps legends and choropleths. For the LST maps, what have the authors defined as low, medium and high correlation (note: some standard definitions exist)? For NDVI, green is more intuitive for high NDVI values (i.e., healthy vegetation). Many Figure map legends must provide the map feature explanation that cartographic practice requires.

In terms of minor adjustments, this reviewer requests the following:

-        The methods section needs reorganising to clarify the methods used – the summary of lines 123 to 128 does not cover later method explanation.

Answer:

Thank you for your comments. This section was revised and modified for more clarity. The section between lines 123 and line 128 was updated and moved to the methodology section for more consistency. Objectives 1, 2 and 3 are presented in the result section and objectives 4 and 5 were supported by the analysis discussing them in depth in the discussion. Now, it reads as follows: 

“The specific objectives of this study are: (1) to retrieve LST from Landsat 8 OLI and TIRS data and prepare spatial LST distribution maps for the holy sites of Makkah; (2) to generate contour lines of the output LST; (3) to identify the UHI and non-UHI, based on the retrieved LST; (4) to map the relation between pedestrian networks and LST differences in each holy sites; and (5) to integrate the UHIs with the pedestrian network in the holy sites.”

-        A fuller explanation for why a correlation analysis of NDVI (etc) with LST is essential for the study. What is the physical basis for that relationship – and will it help indicate in the forthcoming results? The use of NDBI similarly needs much more explanation.

Answer:
Additional explanations and references were added to the discussion section to clarify the importance of NDVI and LST correction in the discussion section. Now, it reads as follows:

“The most widely known and frequently used land cover metric is NDVI, which calculates the percentage of vegetation in a pixel [7, 8]. [9] argued that in desert environments, the UHI can deviate from the standard distribution, where that previously noted from global assessments, more populated areas have relatively higher LST compared to the adjacent countryside [10].”

-        A discussion on Landsat spatial resolution – and the inherent limitations would be helpful.

Answer:

We agree with you. An additional paragraph was added to emphasise the Landsat spatial resolution issue in the discussion section.

“[11] discuss the impact of spatial resolution and temporal resolution on UHI analysis. Although multiple data fusion techniques were applied using different earth observation datasets to estimate LST, Landsat imagery remains the most used dataset due to data availability and as open-source data. However, The usefulness of Landsat spatial resolution with 30 m and other spatial resolution datasets up to 50 m were evaluated and recommended by [11]. Furthermore, the terrain factors affect the LST as approved in this study, and the degree of influence needs further investigation as discussed in [12].”

-        The Figure numbers in the text do not always match the Figure captions.

Answer:

Thank you for your comments. The order of figures in the text has been revised.

Noting these difficulties, the reviewer did not review the contents of the Discussion sections.

Answer:

Thank you for your effort and time. The discussion section was modified based on reviewers` comments to represent the actual work, the used methodology and the significant funding.

Comments on the Quality of English Language

Minor grammatical errors to be corrected - for example, line 15 - change 'increase the number pilgrims' to 'increase the number of pilgrims'.

Answer:
Thank you for your comments. The sentence was modified based on your comments and after English editing too. Now, it reads as follows:

“During Hajj, Muslims throughout the globe assemble in the holy sites of Makkah, Saudi Arabia. The Saudi government aims to increase the number of pilgrims.”

Submission Date

11 July 2023

Date of this review

25 Jul 2023 02:55:53

[1]         S. Gazette. "Entry to holy sites." https://saudigazette.com.sa/article/595641/SAUDI-ARABIA/Entry-to-holy-sites-restricted-from-July-19 (accessed 17 September 2022).

[2]         Reuters. "Muslims begin annual haj pilgrimage in Mecca." https://www.reuters.com/article/us-saudi-haj-arafat-idUSKCN1BA1J7 (accessed 17 October, 2023).

[3]         G. Mutani and V. Todeschi, "The Effects of Green Roofs on Outdoor Thermal Comfort, Urban Heat Island Mitigation and Energy Savings," Atmosphere, vol. 11, no. 2, doi: 10.3390/atmos11020123.

[4]         D. Hidalgo García, "Evaluation and Analysis of the Effectiveness of the Main Mitigation Measures against Surface Urban Heat Islands in Different Local Climate Zones through Remote Sensing," Sustainability, vol. 15, no. 13, p. 10410, 2023. [Online]. Available: https://www.mdpi.com/2071-1050/15/13/10410.

[5]         D. A. Artis and W. H. Carnahan, "Survey of emissivity variability in thermography of urban areas," (in English), Remote Sensing Environ.; (United States), vol. 12, 1982/09/01/ 1982, doi: 10.1016/0034-4257(82)90043-8.

[6]         P. Schmidt and B. T. Lawrence, "Association between Land Surface Temperature and Green Volume in Bochum, Germany," Sustainability, vol. 14, no. 21, p. 14642, 2022. [Online]. Available: https://www.mdpi.com/2071-1050/14/21/14642.

[7]         K. Deilami, M. Kamruzzaman, and Y. Liu, "Urban heat island effect: A systematic review of spatio-temporal factors, data, methods, and mitigation measures," International Journal of Applied Earth Observation and Geoinformation, vol. 67, pp. 30-42, 2018/05/01/ 2018, doi: https://doi.org/10.1016/j.jag.2017.12.009.

[8]         A. Fallatah, S. Jones, L. Wallace, and D. Mitchell, "Combining Object-Based Machine Learning with Long-Term Time-Series Analysis for Informal Settlement Identification," Remote Sensing, vol. 14, no. 5, p. 1226, 2022. [Online]. Available: https://www.mdpi.com/2072-4292/14/5/1226.

[9]         M. Mohamed, A. Othman, A. Z. Abotalib, and A. Majrashi, "Urban Heat Island Effects on Megacities in Desert Environments Using Spatial Network Analysis and Remote Sensing Data: A Case Study from Western Saudi Arabia," Remote Sensing, vol. 13, no. 10, p. 1941, 2021. [Online]. Available: https://www.mdpi.com/2072-4292/13/10/1941.

[10]       Z. Liang et al., "Seasonal and Diurnal Variations in the Relationships between Urban Form and the Urban Heat Island Effect," Energies, vol. 13, no. 22, p. 5909, 2020. [Online]. Available: https://www.mdpi.com/1996-1073/13/22/5909.

[11]       J. A. Sobrino, R. Oltra-Carrió, G. Sòria, R. Bianchi, and M. Paganini, "Impact of spatial resolution and satellite overpass time on evaluation of the surface urban heat island effects," Remote Sensing of Environment, vol. 117, pp. 50-56, 2012/02/15/ 2012, doi: https://doi.org/10.1016/j.rse.2011.04.042.

[12]       X. Peng, W. Wu, Y. Zheng, J. Sun, T. Hu, and P. Wang, "Correlation analysis of land surface temperature and topographic elements in Hangzhou, China," (in en), Sci Rep, vol. 10, no. 1, p. 10451, 2020/06/26/ 2020, doi: 10.1038/s41598-020-67423-6.

[13]       I. N. Sunarta, R. Suyarto, M. Saifulloh, W. Wiyanti, K. D. Susila, and L. G. L. Kusumadewi, "Surface Urban Heat Island (Suhi) Phenomenon In Bali And Lombok Tourism Areas Based On Remote Sensing," Journal of Southwest Jiaotong University, vol. 57, no. 4, 2022.

Reviewer 4 Report

The article deals with UHI pattern in Mecca during Hajj Pilgrimage. The approach is unique and novel and the study can significantly contribute to the UHI research. The article is nicely designed and well written. I have just suggestion and one question.

It will be good to calculate the UHI and other indices for before and after hajj and see how the UHI pattern changes.

Authors have not done Validation of UHI/LST. I recommend authors to do the validation. 

In table 1. Author have mentioned that cloud cover of scene was >30% which is very high and might have significant impact on the accuracy. Is it a mistyping? If not then make it clear that is their any cloud cover over the study region.

Author Response

Response to reviewer 4 comments

General comments

Thank you for your comments and engagements. The manuscript has been updated as per the reviewers' recommendation where possible. We provide a justification for each question highlighting the modifications made in red in the manuscript. We have responded to each criterion separately and modified and corrected the thesis accordingly. Comprehensive editing for the whole manuscript has been completed to increase readability and clarity Using the MDPI English service. Kindly, find the updated references in the manuscript and some references used for justification in our responses at the end of the response template.

Open Review

Quality of English Language

( ) I am not qualified to assess the quality of English in this paper
( ) English very difficult to understand/incomprehensible
( ) Extensive editing of English language required
( ) Moderate editing of English language required
( ) Minor editing of English language required
(x) English language fine. No issues detected

Yes

Can be improved

Must be improved

Not applicable

Is the content succinctly described and contextualized with respect to previous and present theoretical background and empirical research (if applicable) on the topic?

(x)

( )

( )

( )

Are all the cited references relevant to the research?

( )

(x)

( )

( )

Are the research design, questions, hypotheses and methods clearly stated?

(x)

( )

( )

( )

Are the arguments and discussion of findings coherent, balanced and compelling?

(x)

( )

( )

( )

For empirical research, are the results clearly presented?

( )

(x)

( )

( )

Is the article adequately referenced?

(x)

( )

( )

( )

Are the conclusions thoroughly supported by the results presented in the article or referenced in secondary literature?

( )

(x)

( )

( )

Comments and Suggestions for Authors

The article deals with UHI pattern in Mecca during Hajj Pilgrimage. The approach is unique and novel and the study can significantly contribute to the UHI research. The article is nicely designed and well written. I have just suggestion and one question.

Answer:

Thank you for your comments and encouragement. Additionally, the manuscript has been updated as per the reviewers' recommend where possible to increase readability and enhance the consistency.

It will be good to calculate the UHI and other indices for before and after hajj and see how the UHI pattern changes.

Answer:

Thank you for your comments. Performing time series analysis (TSA) to estimate the LST variation in regions using temporal resolution would be a great idea. However, implementing TSA in this study is outside the scope of this study. This suggestion can be a future work to represent UHI variation before, during and after the Hajj season. We applied the method in three sites to investigate LST correlation with the digital elevation model and remote sensing indices. This work is generic and can be applied in similar urban and environmental contexts. However, this idea was introduced briefly when detecting the mitigation solution was introduced to the area in the discussion section.   

Authors have not done Validation of UHI/LST. I recommend authors to do the validation. 

Answer:

Thank you for your comments. Since Landsat data is only used for LST analysis, the correlation between LST results and remote sensing indicators such as NDVI, NDBI and the elevation model can be used to verify the result in multiple dimensions. These correlations were interpreted quantitively using a scatter diagram as shown in the discussion section. Some researchers applied different techniques using image fusion to compare the result from different datasets such as MODIS and Landsat [13]. However, this fusion affects the LST analysis as discussed in [4]. Therefore,  following [4, 9] the authors decided to use just Landsat data.

In table 1. Author have mentioned that cloud cover of scene was >30% which is very high and might have significant impact on the accuracy. Is it a mistyping? If not then make it clear that is their any cloud cover over the study region.

Answer:

Table 1 was updated and re-paraphrased to indicate there was 0% cloud cover over the study region. >30 % was used as the main criteria to download the images. However, the obtained images have free cloud cover during the summer period. 

[1]         S. Gazette. "Entry to holy sites." https://saudigazette.com.sa/article/595641/SAUDI-ARABIA/Entry-to-holy-sites-restricted-from-July-19 (accessed 17 September 2022).

[2]         Reuters. "Muslims begin annual haj pilgrimage in Mecca." https://www.reuters.com/article/us-saudi-haj-arafat-idUSKCN1BA1J7 (accessed 17 October, 2023).

[3]         G. Mutani and V. Todeschi, "The Effects of Green Roofs on Outdoor Thermal Comfort, Urban Heat Island Mitigation and Energy Savings," Atmosphere, vol. 11, no. 2, doi: 10.3390/atmos11020123.

[4]         D. Hidalgo García, "Evaluation and Analysis of the Effectiveness of the Main Mitigation Measures against Surface Urban Heat Islands in Different Local Climate Zones through Remote Sensing," Sustainability, vol. 15, no. 13, p. 10410, 2023. [Online]. Available: https://www.mdpi.com/2071-1050/15/13/10410.

[5]         D. A. Artis and W. H. Carnahan, "Survey of emissivity variability in thermography of urban areas," (in English), Remote Sensing Environ.; (United States), vol. 12, 1982/09/01/ 1982, doi: 10.1016/0034-4257(82)90043-8.

[6]         P. Schmidt and B. T. Lawrence, "Association between Land Surface Temperature and Green Volume in Bochum, Germany," Sustainability, vol. 14, no. 21, p. 14642, 2022. [Online]. Available: https://www.mdpi.com/2071-1050/14/21/14642.

[7]         K. Deilami, M. Kamruzzaman, and Y. Liu, "Urban heat island effect: A systematic review of spatio-temporal factors, data, methods, and mitigation measures," International Journal of Applied Earth Observation and Geoinformation, vol. 67, pp. 30-42, 2018/05/01/ 2018, doi: https://doi.org/10.1016/j.jag.2017.12.009.

[8]         A. Fallatah, S. Jones, L. Wallace, and D. Mitchell, "Combining Object-Based Machine Learning with Long-Term Time-Series Analysis for Informal Settlement Identification," Remote Sensing, vol. 14, no. 5, p. 1226, 2022. [Online]. Available: https://www.mdpi.com/2072-4292/14/5/1226.

[9]         M. Mohamed, A. Othman, A. Z. Abotalib, and A. Majrashi, "Urban Heat Island Effects on Megacities in Desert Environments Using Spatial Network Analysis and Remote Sensing Data: A Case Study from Western Saudi Arabia," Remote Sensing, vol. 13, no. 10, p. 1941, 2021. [Online]. Available: https://www.mdpi.com/2072-4292/13/10/1941.

[10]       Z. Liang et al., "Seasonal and Diurnal Variations in the Relationships between Urban Form and the Urban Heat Island Effect," Energies, vol. 13, no. 22, p. 5909, 2020. [Online]. Available: https://www.mdpi.com/1996-1073/13/22/5909.

[11]       J. A. Sobrino, R. Oltra-Carrió, G. Sòria, R. Bianchi, and M. Paganini, "Impact of spatial resolution and satellite overpass time on evaluation of the surface urban heat island effects," Remote Sensing of Environment, vol. 117, pp. 50-56, 2012/02/15/ 2012, doi: https://doi.org/10.1016/j.rse.2011.04.042.

[12]       X. Peng, W. Wu, Y. Zheng, J. Sun, T. Hu, and P. Wang, "Correlation analysis of land surface temperature and topographic elements in Hangzhou, China," (in en), Sci Rep, vol. 10, no. 1, p. 10451, 2020/06/26/ 2020, doi: 10.1038/s41598-020-67423-6.

[13]       I. N. Sunarta, R. Suyarto, M. Saifulloh, W. Wiyanti, K. D. Susila, and L. G. L. Kusumadewi, "Surface Urban Heat Island (Suhi) Phenomenon In Bali And Lombok Tourism Areas Based On Remote Sensing," Journal of Southwest Jiaotong University, vol. 57, no. 4, 2022.

Round 2

Reviewer 2 Report

Thanks for addressing the review comments. 

Author Response

Thank you for your comments 
The manuscript was modified based on reviewers' comments 
Regard

Reviewer 3 Report

I have provided a comment to the journal editors.

Author Response

Response to reviewer 3 and editorial office comments

General comments

Thank you for your comments and engagements. The manuscript has been updated as per the reviewer's recommendation where possible. We provide a justification for each question highlighting the modifications made in red in the manuscript. We have responded to each criterion separately and modified and corrected the thesis accordingly. A comprehensive editing for the whole manuscript has been completed to increase readability and clarity Using the MDPI English service. Kindly, find the updated references in the manuscript and some references used for justification in our responses at the end of the response template.

Comments and Suggestions for Authors

Reviewer 3 gave the following suggestions after reviewing the revised
manuscript: "In my view, the authors have tinkered on the edges of
correcting the paper  - but (for me) the substantive issue remains. I am
of the opinion that the use of Gaussian statistics has not been applied
here appropriately."
Answer:

We would like to thank you for your comment. Here, we would like to address two points:

Firstly, the used figures such as figure numbers 8,10-12, illustrate the selected sections across pedestrian paths to show LST profiles using the correlation graph. The LST profile distribution represents the real reflection of the used materials in this environment. The section comes across different materials and elevation to investigate the correlation between LST and Elevation.

Secondly, the calculation of LST takes five steps. In the methodology section, the base of this calculation was clearly mentioned in the second paragraph under Table 1. Additionally, the statement can be found at the end of the paragraph as follows:

 “Following [1-4], the calculation procedure was conducted as shown in Table2.”

An additional statement was added to clarify the correlation graph approach as follows:  The information of the correlation between LST and other parameters can be found in detail in [4, 5].”

After comparing in the database, we found some contents in your
manuscript are the same with those in the previous works (Source 1). We
have attached the detected report here. To keep the originality for your
manuscript, we suggest you rewrite these parts. Please note self-reuse
is not allowed for a scientific article either. If possible, please
detect it in the database after rewriting.

Answer:

Thank you for your comments. The copyright issue was modified via a comprehensive editing of the manuscript. In the fact, some of the points in the originality reports maybe needs to be reconsidered such as affiliation, some statement in the MDPI template and bibliography. Therefore, it is impossible to change. However, we are grateful for these comments, which can clarify and enhance our work.

.

________________________________________
References

[1]         D. A. Artis and W. H. Carnahan, "Survey of emissivity variability in thermography of urban areas," (in English), Remote Sensing Environ.; (United States), vol. 12, 1982/09/01/ 1982, doi: 10.1016/0034-4257(82)90043-8.

[2]         D. Hidalgo García, "Evaluation and Analysis of the Effectiveness of the Main Mitigation Measures against Surface Urban Heat Islands in Different Local Climate Zones through Remote Sensing," Sustainability, vol. 15, no. 13, p. 10410, 2023. [Online]. Available: https://www.mdpi.com/2071-1050/15/13/10410.

[3]         P. Schmidt and B. T. Lawrence, "Association between Land Surface Temperature and Green Volume in Bochum, Germany," Sustainability, vol. 14, no. 21, p. 14642, 2022. [Online]. Available: https://www.mdpi.com/2071-1050/14/21/14642.

[4]         D. H. García, H. Rezapouraghdam, C. M. Hall, O. M. Karatepe, and S. N. Koupaei, "Spatio-temporal variability of the earth’s surface temperature and the changes in land user/land cover: implications for sustainable tourism development," Journal of Policy Research in Tourism, Leisure and Events, pp. 1-28, 2023, doi: 10.1080/19407963.2023.2242362.

[5]         S. Guha, H. Govil, A. Dey, and N. Gill, "Analytical study of land surface temperature with NDVI and NDBI using Landsat 8 OLI and TIRS data in Florence and Naples city, Italy," European Journal of Remote Sensing, vol. 51, no. 1, pp. 667-678, 2018.

Round 3

Reviewer 3 Report

I have nothing further to add from my previous review.